How many fish? Comparison of two underwater visual sampling methods for monitoring fish communities

Thanopoulou Zoi 1 2 zxt89@miami.edu
Sini Maria 3
Vatikiotis Konstantinos 3
Katsoupis Christos 3
http://orcid.org/0000-0002-8374-4392 Dimitrakopoulos Panayiotis G. 2
http://orcid.org/0000-0002-5137-7540 Katsanevakis Stelios 3
1 Department of Biology, University of Miami , Miami, FL , USA
2 Department of the Environment, Aegean University , Mytilene , Greece
3 Department of Marine Sciences, Aegean University , Mytilene , Greece
Kramer Donald
Electronic publication date: 2018 Jun 20
Publication date: 2018
Volume: 6
Electronic Location ID: e5066
Received 2018 Jan 23; Accepted 2018 Jun 3
Copyright: © 2018 Thanopoulou et al.
Copyright year: 2018
Copyright holder: Thanopoulou et al.
License: This is an open access article distributed under the terms of the Creative Commons Attribution License, which permits unrestricted use, distribution, reproduction and adaptation in any medium and for any purpose provided that it is properly attributed. For attribution, the original author(s), title, publication source (PeerJ) and either DOI or URL of the article must be cited.
License URL: https://creativecommons.org/licenses/by/4.0/

Keywords: Underwater visual census, Line transects, Strip transects, Rocky reefs, Bias, Mediterranean, Belt transects

Funding: DG for Maritime Affairs and Fisheries of the European Commission SI2.721917 This work was supported by the Research Project “PROTOMEDEA—Toward the establishment of Marine Protected Area Networks in the Eastern Mediterranean,” funded by DG for Maritime Affairs and Fisheries of the European Commission, under Grant Agreement SI2.721917. The funders had no role in study design, data collection and analysis, decision to publish, or preparation of the manuscript.

==============================
Background

Underwater visual surveys (UVSs) for monitoring fish communities are preferred over fishing surveys in certain habitats, such as rocky or coral reefs and seagrass beds and are the standard monitoring tool in many cases, especially in protected areas. However, despite their wide application there are potential biases, mainly due to imperfect detectability and the behavioral responses of fish to the observers.

Methods

The performance of two methods of UVSs were compared to test whether they give similar results in terms of fish population density, occupancy, species richness, and community composition. Distance sampling (line transects) and plot sampling (strip transects) were conducted at 31 rocky reef sites in the Aegean Sea (Greece) using SCUBA diving.

Results

Line transects generated significantly higher values of occupancy, species richness, and total fish density compared to strip transects. For most species, density estimates differed significantly between the two sampling methods. For secretive species and species avoiding the observers, the line transect method yielded higher estimates, as it accounted for imperfect detectability and utilized a larger survey area compared to the strip transect method. On the other hand, large-scale spatial patterns of species composition were similar for both methods.

Discussion

Overall, both methods presented a number of advantages and limitations, which should be considered in survey design. Line transects appear to be more suitable for surveying secretive species, while strip transects should be preferred at high fish densities and for species of high mobility.

Introduction

In recent decades, several sampling approaches have been developed for the assessment of fish communities in different marine habitat types. Selecting the most suitable sampling method is a crucial step during the survey design process. This decision is usually dictated by the overall research objectives, the level of accuracy needed to address scientific questions, the time and resource availability to carry out the survey, as well as the physical, ecological, and behavioral characteristics of the fish and habitats under investigation (Lessios, 1996; Rotherham et al., 2007). Non-destructive approaches such as underwater visual survey (UVS) methods (Hill & Wilkinson, 2004; Andaloro et al., 2011, 2013) are preferred when assessing protected or endangered species or when sampling in vulnerable habitat types such as coral reefs and seagrass meadows.

Underwater visual survey methods for the assessment of fish include five main quantitative or semi-quantitative methods, which can be carried out either through SCUBA or free diving or through the examination of video and photographic records. These methods include plot sampling (strip transects and point counts), distance sampling (line transects and point transects), fixed-time transects, occupancy estimation based on repetitive sampling, and rapid visual techniques (described in detail in Katsanevakis et al., 2012). In this study, the first two methods (specifically, strip transects and line transects) were further analyzed and compared. These two methods were selected as they are the only UVS techniques that provide absolute abundance estimations, while the other three are less informative, as they provide estimates of either indices of abundance or probability of presence.

In shallow water reef fish assemblages, plot sampling, and especially the strip transect method, is the most widely used UVS technique (Samoilys & Carlos, 2000; Caldwell et al., 2016; Friedlander et al., 2018). Strip transects are a simple, low-cost technique that can be performed through SCUBA diving or snorkeling, depending on water visibility and depth, with minimum equipment requirements (Holmes et al., 2013). During strip transects, observations of target fish are made within a predetermined surface area (Côté & Perrow, 2006). Mapstone & Ayling (1993) proposed that mid-sized strips, that is, 50 or 75 m length and 5 or 10 m width, are suitable to obtain a representative sample of the fish community. The optimal swimming speed of the observer is usually accepted to be a compromise between a rapid constant pace (necessary to avoid implications due to fish movement) and search efficiency (Samoilys & Carlos, 2000).

A crucial assumption in strip transect sampling is that detectability within the investigated area is perfect. Yet again, when assessing fish populations, there are several reasons that may lead to imperfect detectability, and subsequently, result in an underestimation of species composition and population density (Monk, 2014). Several studies have shown that detectability varies considerably across fish species and is mostly affected by body size, schooling behavior, shyness, and secretive coloration or behavior (MacNeil et al., 2008b; Bozec et al., 2011). Environmental factors such as habitat complexity (Edgar & Barrett, 1999) and water visibility (MacNeil et al., 2008a, 2008b) also influence detectability. Alongside the various morphological and ecological characteristics of different species and habitats, several methodological factors, such as the selection of strip width, also affect the level of detectability (Kulbicki & Sarramégna, 1999). Consequently, species richness and abundance may be substantially underestimated in strip transects (Franzreb, 1981; Katsanevakis, 2009).

In many cases, the problem of imperfect detectability can be addressed through distance sampling, as this method accounts for detection probability (Buckland et al., 2001, 2004). In the marine environment, line transects are the most commonly used distance sampling technique (Katsanevakis et al., 2012). The sampling process is similar to that of strip transects, but fish observations are not restricted within a pre-defined strip width; instead, the perpendicular distance of each fish observation from the transect line is recorded. These perpendicular distances are then used to account for the detection probability (Buckland et al., 2001). Estimating the detection probability (Pa) is the most important task of the analysis related to distance sampling data (Burnham & Anderson, 1984; Buckland et al., 2001).

A critical assumption of distance sampling that should be ensured by the survey design and protocols is that detection on or near the line is perfect (Buckland et al., 2001; Thomas et al., 2010). In the case of violation of this assumption, a negative bias in the estimation of abundance is expected. Another important requirement is that all measurements of the distances are precise. Tape lines and laser rangefinders usually offer more precise measurements than rough estimates by eye, which may be affected by the observer’s visual ability (Thresher & Gunn, 1986). Moreover, water turbidity may also affect distance estimations, as in clear waters distances are commonly underestimated, while in turbid waters they are overestimated (Kulbicki, 1998).

Both methods suffer from many additional sources of bias. They both depend on the observer’s ability to identify fish species in situ (Thompson & Mapstone, 1997). Fish are assumed to be observed at their original location, before being influenced by the researcher’s presence, as important bias in abundance estimation may be caused due to fish movement in response to observer’s presence (Fewster et al., 2008). This largely depends on the behavior of different fish species; if individuals are attracted by the researcher the bias will be positive, while in the case of avoidance, the bias will be negative. Abundance will also be overestimated if the same individuals are recorded more than once due to their movement ahead of the observer. Biases caused by the observer are likely to be restricted when the observer is experienced (Sale & Sharp, 1983; Thompson & Mapstone, 1997), while biases related to the distribution and behavior of individuals will differ according to the field protocols. The response of fish toward the observer also varies according to the different levels of fishing pressure in the area under study (Bohnsack & Bannerot, 1986; Bellwood, 1998). Kulbicki (1998) showed that, due to divergent fish behavior, marine protected areas would seem to have higher estimated fish densities than areas with high fishing pressure even for the same real values of density.

Other recent studies have examined and/or compared the performance of various UVS methods (Bosch et al., 2017; Irigoyen et al., 2018). Bosch et al. (2017) compared the output of three different methods for studying fish assemblages, UVC (strip transects of fixed length and width), Fish Traps, and Baited Cameras. The authors mainly focused on species diversity; they did not assess the performance of the methods to estimate species absolute or relative abundance. Irigoyen et al. (2018) used conventional strip transects of various fixed width and the distance sampling technique in combination with the “Tracked Roaming Diver” technique (a sampling method that maximizes the length of the transect and thus the area studied). They proposed that the combination of the two latter methods provides a more efficient way for UVS. However, in their study, only six species of the Western Mediterranean Sea were analyzed restraining their analysis and conclusions to only commercially important, medium- to large-sized reef fishes of the specific area. Comparisons between strip and line transects for UVS for fish have been previously conducted but either focused on specific fish families or on single state metrics, most commonly on population density (Thresher & Gunn, 1986; Kulbicki & Sarramégna, 1999).

The aim of our study was to quantitatively compare the performance of the strip and line transect methods, for the assessment and monitoring of Mediterranean rocky reef fish in a non-destructive manner. We estimated a number of univariate and multivariate metrics, such as species occupancy, richness, population density, and composition. For practical and logistical reasons, the study focused on a representative pre-selected group of twenty rocky reef demersal species covering the widest possible range of trophic groups and behaviors and including both commercially important and non-important species.

Methodology

Study area

The study area comprises the Greek territorial waters of the Aegean Sea. The study was conducted from July to October 2016 and included 31 rocky reef sites (Fig. 1). None of the sites was in a no-take zone; similar general fishing restrictions applied to all sites with a few exceptions of increased restrictions (Petza et al., 2017). At all but one site (due to lack of appropriate substratum), two stations were surveyed. There was a minimum distance of 50 m between transects of the two stations at each site.

Figure 1 Map of the sampling area depicting the different sites and the code numbers of sampling stations.

The inset depicts the study area within the Mediterranean Sea.

Sampling methods and target species

At every station both strip and line transects on rocky reef habitats were surveyed by SCUBA diving. The exact location of the starting point of each strip transect was randomly selected at a depth between 5 and 15 m; the transect followed an approximately constant depth contour. The starting point of the paired line transect was placed ∼10 m away, and the line transect was conducted in the opposite direction. All surveys were conducted between 10 am and 4 pm, and in every case underwater visibility was at least 20 m. The same set of four observers conducted all surveys. All strip transects were 75 m long and 5 m wide (2.5 m on either side of the transect line). In order to minimize disturbance, fish recording, and transect deployment were done simultaneously by the observers. Line transects were also 75 m long; the perpendicular distances of individual fish (or groups of fish) from the line at time of first detection were measured using a measuring tape and a fixed point on the substrate as a reference point, for fish detected up to 8 m on either side of the central line. For this purpose, two divers worked together; the first deployed the transect line and held the end of the measure tape ensuring that it remained vertical to the transect line, while the second diver counted the fish and measured the perpendicular distance. When individuals were observed in groups, the distance of the center of the group was estimated as well as the number of individuals. The swimming speed of the observer for the strip transect method was approximately 3.1 m/min, while the corresponding speed for the line transect method was 2.2 m/min. The survey targeted 20 fish taxa (Table 1).

Table 1 Fish taxa surveyed (according to Horton et al., 2018).

Family	Species	Authority	
Moronidae	Dicentrachus labrax	Linnaeus (1758)	
Mullidae	Mullus surmuletus	Linnaeus (1758)	
Muraeninae	Muraena helena	Linnaeus (1758)	
Scaridae	Sparisoma cretense	Linnaeus (1758)	
Scianidae	Sciaena umbra	Linnaeus (1758)	
Scorpaenidae	Scorpaena spp.	Linnaeus (1758)	
Serranidae	Epinephelus costae	Steindachner (1878)	
Serranidae	Epinephelus marginatus	Lowe (1834)	
Serranidae	Serranus cabrilla	Linnaeus (1758)	
Serranidae	Serranus scriba	Linnaeus (1758)	
Siganidae	Siganus luridus	Rüppell (1829)	
Siganidae	Siganus rivulatus	Forsskål & Niebuhr (1775)	
Sparidae	Dentex dentex	Linnaeus (1758)	
Sparidae	Diplodus annularis	Linnaeus (1758)	
Sparidae	Diplodus puntazzo	Walbaum (1792)	
Sparidae	Diplodus sargus	Linnaeus (1758)	
Sparidae	Diplodus vulgaris	Geoffroy Saint-Hilaire (1817)	
Sparidae	Oblada melanura	Linnaeus (1758)	
Sparidae	Sarpa salpa	Linnaeus (1758)	
Sparidae	Spondyliosoma cantharus	Linnaeus (1758)	
Note:

Scorpaena spp. include the species Scorpaena porcus, Scorpaena scrofa, and Scorpaena notata, which cannot be easily distinguished in situ.

Estimating population densities

In strip transects, the population mean density was estimated by the formula:D^=n2wL=nAc

n, number of individuals; 2w, total width of the transect; L, length of the transect; Ac, total covered (sampled) area.

Bootstrap (bias-corrected and accelerated with 1,000 permutations) was applied to estimate, for each species, the unconditional standard error (Efron & Tibshirani, 1993), as well as the 95% bootstrap-based unconditional confidence interval of the mean density, using R version 3.2.3 (R Core Team, 2015).

For line transect data, the mean density was estimated by:D^=n(AcPa)

where Pa is the detection probability, given by:Pa=∫0wg(y)dyw

where w is the half-width of the line transects and g(y) is the detection function, representing the probability of detecting an individual that is at a distance y from the transect line (Buckland et al., 2001).

The function g(y) was estimated from the distance data (grouped data, right truncated at width that varied from 1.2 to 8 m, depending on the dataset of each species to exclude outliers) with a semi-parametric approach, according to Buckland et al. (2001), using the software DISTANCE 6.2 (Thomas et al., 2010). Specifically, the detection function was modeled in the general form:g(y)=key(y)[1+series(y)]key(0)[1+series(0)]

where key(y) is the key function and series(y) is a series expansion used to adjust the key function. The uniform function key(y) = 1/w (0 parameters), the one parameter half normal function key(y)=exp(−y22σ2) and the two-parameter hazard-rate function key(y)=1−exp[−(yσ)−b] were considered as key functions; three series expansions were considered: the cosine series ∑j=1majcos(jπy/W), simple polynomials of the form ∑j=1maj(yW)2j and hermite polynomials of the form ∑j=2majH2j(y/σ), where σ and aj are the best-fit parameters (Buckland et al., 2001).

A total of six models were considered for g(y): uniform key with cosine or simple polynomial series expansions, the half normal key with cosine or hermite polynomial series expansions and hazard-rate key with cosine or simple polynomial series expansions, as proposed by Buckland et al. (2001). Model selection was based on the Akaike’s information criterion (AIC) (Akaike, 1973). The number j of parameters in each series expansion was also defined using AIC between models of increasing order. The model with the smallest AIC value (AICmin) was selected as the “best” among the models tested.

Comparing occupancy, species richness and density estimates between strip and line transects

The occupancy of each species (percentage of stations in which the species were recorded), species richness and population density estimates, based on the two different sampling methods were compared. Occupancy was estimated for each of the 20 species per method separately. This resulted into two distinct datasets, each consisting of 20 occupancy values; one for the line transect method and one for the strip transects method. The set of differences between line and strip transects (i.e., line transects minus strip transects) was then subjected to bootstrapping, to estimate the mean value and 95% confidence interval of the differences.

Similarly, the comparison of species richness values obtained by the two different methods (i.e., number of species present among the 20 target species) was achieved through the bootstrapping technique. Initially, species richness was estimated for each station and method separately. Consequently, two datasets of 61 species richness values each were obtained for the two methods. The set of differences when subtracting the second dataset from the first, was the actual dataset that was bootstrapped.

A similar procedure was followed for the comparison of the density estimates between the two methods. For the comparison of the “overall densities,” the mean density of each species over all stations was estimated by each method, and the differences between the two datasets (comprising of the 20 mean densities of distinct species) were bootstrapped. Additionally, the density for each species at each station was also estimated. Therefore, for each species two datasets (one for each method) with 61 values, corresponding to the number of stations, were created. The differences by subtracting the dataset of strip transects from the dataset of line transects were bootstrapped to estimate the confidence interval of the differences and test if it differed from zero. Stations in which a species was not recorded were excluded from the analysis of that species, as the aim was to test for differences in the estimates of densities between the two methods when a species was actually present (as inferred by at least one of the methods).

Species composition

To investigate potential differences in species composition between the two sampling methods, a Bray–Curtis similarity matrix was generated based on a square-root transformation of fish density data, which was then used to carry out cluster analysis and construct a non-metric multidimensional scaling plot. In this case, fish density data (by both methods) derived only from one of the two stations of each site (31 stations in total) were used, in order to improve clarity. Otherwise, the resulting MDS plot and dendrogram were too crowded (with 122 points—61 stations × 2 methods). Moreover, in the respective plots different colors were used for the visual depiction of the station geographical position; stations marked with cold colors (shades of blue) refer to areas of the northern Aegean, stations marked with warm colors (yellow/orange/red) are located in the southern Aegean, while green colors denote stations found in the central Aegean Sea. A SIMPER analysis based on all stations was conducted to identify the species that contributed most to the observed variability between the two sampling methods. The species composition analysis was carried out with PRIMER 6 software (Clarke & Gorley, 2006).

Results

Distance sampling analysis

For each species, the best model, based on AIC, was used for inference (Table 2). An empirical minimum of observations to model the detection function is 30 observations (Buckland et al., 1993). However, a number of species did not fulfill this requirement. These species were Dentex dentex, Epinephelus marginatus, Muraena helena, Sciaena umbra, and Spondyliosoma cantharus. The highest detectability values (excluding species with very low number of observations <30) were recorded for E. costae (detectability value ± SE, 0.84 ± 0.15) followed by Siganus luridus (0.73 ± 0.03). The lowest detectability values were recorded for Scorpaena spp. (0.32 ± 0.05) and Serranus cabrilla (0.41 ± 0.07) followed by Mullus surmuletus (0.49 ± 0.06). The estimated detection probability curves corresponded to different fish behaviors (Fig. 2), following the terminology of Kulbicki (1998). Diplodus annularis, D. puntazzo, D. sargus, D. vulgaris, Oblada melanura, Sparisoma cretense, Siganus luridus, and S. rivulatus exhibited “shy behavior,” that is, avoided the observer. M. surmuletus, E. costae, Serranus scriba, S. cabrilla, and Sarpa salpa had neutral behavior, while Scorpaena spp. were secretive showing a rapid decrease in detectability within the first 0.4 m (Fig. 2A).

Table 2 Best fit model, maximum width of line transect after truncation (w) and value of detectability (Pa) of the DISTANCE analysis for each species.

Species	Model	Wmax (m)	Pa (SE)	
Disentrachus labrax	–	–	–	
Mullus surmuletus	Hazard rate, simple polynomial of order 2	8.0	0.49 ± 0.06	
Muraena helena*	Half normal, cosine of order 1	4.2	0.99 ± 0.72	
Sparisoma cretense	Hazard rate, simple polynomial of order 2	6.0	0.79 ± 0.03	
Sciaena umbra*	Half normal, cosine of order 1	1.4	0.99 ± 0.75	
Scorpaena spp.	Half normal, cosine of order 2	1.2	0.32 ± 0.05	
Epinephelus costae	Hazard rate, hermite of order 2	6.5	0.84 ± 0.15	
Epinephelus marginatus*	Uniform, cosine of order 1	7.0	0.58 ± 0.09	
Serranus cabrilla	Hazard rate, simple polynomial of order 1	5.0	0.41 ± 0.07	
Serranus scriba	Half normal, hermite of order 1	6.0	0.54 ± 0.04	
Siganus luridus	Hazard rate, simple polynomial of order 2	6.0	0.73 ± 0.03	
Siganus rivulatus	Uniform, cosine of order 1	6.3	0.56 ± 0.05	
Dentex dentex*	Uniform	7.3	1.00 ± 0.48	
Diplodus annularis	Hazard rate, simple polynomial of order 2	6.9	0.57 ± 0.04	
Diplodus puntazzo	Uniform, cosine of order 1	6.9	0.60 ± 0.04	
Diplodus sargus	Hazard rate, cosine of order 2	6.8	0.64 ± 0.09	
Diplodus vulgaris	Uniform, cosine of order 2	7.0	0.66 ± 0.04	
Oblada melanura	Hazard rate, simple polynomial of order 2	7.6	0.66 ± 0.04	
Sarpa salpa	Hazard rate, simple polynomial of order 2	6.0	0.67 ± 0.05	
Spondyliosoma cantharus*	Uniform	6.7	1.00 ± 0.38	
Notes:

Best fit model, maximum width of line transect after truncation (w) and value of detectability along with the SE (Pa ± SE) of the DISTANCE analysis for each species.

For species marked with “*,” reasonable results were not obtained due to lack of sufficient observations.

Figure 2 Detection probability curves (lines, left ordinate) estimated for three species recorded in line transects illustrating various fish behaviors.

Detection probability curves (lines, left ordinate) estimated for three species recorded in line transects illustrating (A) secretive behavior (Scorpaena spp.), (B) shy behavior (D. annularis), and (C) neutral behavior (E. costae). Bars (right ordinate) represent the frequency of observations in equal distance classes. The numbers above the bars indicate the number of recorded individuals at each distance class.

Species occupancy

Across all sites, D. vulgaris was the most commonly occurring species, as it was recorded in 58 stations by both methods, while Dicentrarchus labrax was never recorded (Fig. 3). Occupancy estimates for the target species varied between the two methods; line transects gave higher estimates in 12 species, strip transects gave higher estimates in four species, while for three species they gave the same estimates (Fig. 3). The highest observed difference was for Scorpaena spp., with an estimated occupancy of 0.64 by line transect sampling and 0.10 by strip transect sampling. The bootstrap method, conducted to compare occupancy estimates (expressed in percentages) between the two methods, showed that overall occupancy was significantly higher when estimated by the line transect method (mean difference: 5.7%; 95% CI [1.3%–11.3%]).

Figure 3 Occupancy estimates per species and per method, expressed as the total number of stations in which the species was present.

Occupancy estimates per species and per method, expressed as the total number of stations in which the species was present (numbers next to the bars), based on surveys by the line (black color/top bar) and strip (blue color/bottom bar) transects methods. The last two bars indicate the mean estimated occupancy of all species per method. The total number of stations was 61.

Species richness

Species richness (i.e., the number of species per station) was estimated to be significantly higher in 36 stations by line transects and in 11 stations by strip transects, while in 14 stations no significant differences were detected between the two methods (Fig. 4). The mean species richness (among the 20 target species) estimated by the line transect method was 8.8, while the corresponding mean species richness estimated by the strip transect method was 7.6. According to the bootstrap method, the mean difference of species richness was 0.98 species [CI: 0.57–1.40], thus indicating significantly higher species richness estimates in line transects than strip transects.

Figure 4 Histogram of the differences in estimated species richness by the line and strip transects methods.

For the calculation, strip transect richness was subtracted from line transect.

Density

Fish density (i.e., number of individuals per hectare) was highly variable both among species and between methods. The overall density (i.e., mean fish density of all species) was higher for line transects than for strip transects, with a value of 166.3 and 119.0 correspondingly. The most abundant species was D. vulgaris, with an estimated mean value of 702.9 individuals per hectare by line transects and 567.8 individuals per hectare by strip transects. Other species with high density were S. salpa, O. melanura, and S. luridus (Table 3, Fig. 5). D. labrax was not found in any station, and the least abundant species, among those present, was S. umbra with a mean density of 1.56 individuals per hectare as estimated by line transects, while no individuals were recorded in the strip transects. Other species with low density were M. helena, D. dentex, and E. marginatus (Table 3, Fig. 5).

Table 3 Mean population densities and 95% confidence intervals for all species per sampling method (line or strip transects).

Species	Method	
Line	Strip	
Mean (individuals/ha)	95% CI	Mean (individuals/ha)	95% CI	
Disentrachus labrax	0	0		0	0	
Mullus surmuletus	48.5	35.3–62.5		44.9	31.0–61.2	
Muraena helena	2.3	1.1–4.1		0.8	0.0–1.7	
Sparisoma cretense	252.3	192.5–317.1		243.0	177.4–312.6	
Sciaena umbra	1.6	0.0–3.9		0	0	
Scorpaena spp.	178.0	127.4–234.1		4.3	1.3–7.8	
Epinephelus costae	13.1	6.1–20.8		21.3	10.0–33.6	
Epinephelus. marginatus	4.8	2.5–7.6		5.2	1.7–10.4	
Serranus cabrilla	85.2	60.7–110.3		63.4	44.1–85.6	
Serranus scriba	232.1	184.2–282.5		167.1	129.3–208.9	
Siganus luridus	529.9	380.5–662.3		281.4	198.0–372.5	
Siganus rivulatus	189.1	102.3–281.3		40.7	15.7–69.5	
Dentex dentex	8.5	1.8–17		1.2	0.0–3.0	
Diplodus annularis	100.7	72.5–132.7		87.8	55.9–124.6	
Diplodus puntazzo	39.0	26.7–53.1		33.5	23.6–44.1	
Diplodus sargus	158.5	121.8–196.7		72.5	56.8–89.1	
Diplodus vulgaris	703.5	606.9–803		568.6	472.5–672.3	
Oblada melanura	312.8	241.5–382.2		319.7	197.5–456.4	
Sarpa salpa	421.1	321.4–525.3		381.7	290.2–478.7	
Spondyliosoma cantharus	3.5	1.6–5.8		40.0	11.8–79.1	
Overall density	166.3	1.5–534.0		119.0	0.0–391.8	

Figure 5 Mean differences of density estimates for each species (with confidence intervals).

Mean differences of density estimates for each species. The bars depict the 95% confidence intervals. The numbers above each point represent the sample size (i.e., the number of stations where the species was present with any of the two methods, when the overall sample size is 61 stations).

The mean difference of the overall fish density was significantly higher for line transects than for strip transects (50.5 individuals per hectare; CI [18.0–85.7]). However, results for individual species varied (Fig. 5). For this calculation, for each species only the stations for which the species was detected by at least one of the methods were included. For D. sargus, D. vulgaris, D. dentex, Scorpaena spp., S. cabrilla, S. scriba, S. luridus, and S. rivulatus, the line transects estimates were significantly higher than strip transects, while the opposite was found for E. costae and S. cantharus. No statistically significant differences between the two methods were found for D. annularis, D. puntazzo, E. marginatus, M. surmuletus, M. helena, O. melanura, S. cretense, and S. salpa. No comparison was possible D. labrax and S. umbra due to lack of data.

Comparing species composition between sampling methods

All data pooled, the two methods presented similar species composition, with an average similarity between methods at each station of 62%. Of the 31 stations presented in Figs. 6 and 7, stations 13, 27, and 15 (indicated by circles in Figs. 6 and 7) presented the highest resemblance (83%, 81.2%, and 81%, respectively) whereas stations 42, 48, and 9 (indicated by lines in Fig. 6 and arrows in Fig. 7) displayed the lowest similarity (i.e., 49.6%, 48%, and 44%, respectively). A total of 12 species contributed the most to the overall differences observed between the two methods (Table 4). Of these S. luridus, S. salpa, O. melanura, and D. vulgaris, S. cretense and S. scriba accounted for approximately 60% of the differences. The observed variability in species composition between the two methods may partly be due to the methods per se and partly due to the between-transect variability at each station. Despite the between-transect variability, a clear separation between distinct geographical regions (North and South Aegean Sea) was obvious in both methods, indicating that both were consistent in depicting large-scale biogeographical patterns.

Figure 6 Two dimensional non-metric multidimensional scaling ordination (MDS) for 31 paired-by-method stations, based on square root-transformation density data and a Bray–Curtis similarity matrix.

Numbers correspond to the stations presented in Fig. 1. Paired-by-method stations with the highest similarities between the two methods are indicated by a circle, while those stations with the lowest similarities are joined with a straight line. North Aegean region stations (1–31 and 56–60) are marked in blue–green colors, while South Aegean stations (33–54) are depicted by red–orange colors.

Figure 7 Cluster analysis of the paired-by-method stations’ similarity, based on square root-transformation density data and a Bray–Curtis similarity matrix.

Numbers correspond to the stations presented in Fig. 1. Paired-by-method stations with the highest similarities are indicated by a circle, while those with the lowest similarities are indicated by arrows of different color. North Aegean region stations (1–31 and 56–60) are marked in blue–green colors, while South Aegean stations (33–54) are depicted by red–orange colors).

Table 4 Summary of similarity percentage analysis (SIMPER) listing species that cumulatively contribute 90% to the dissimilarity (Bray Curtis) of the two underwater visual sampling methods based on square-root transformed density data.

Species	Line transect	Strip transect	Cum.%	
Average dissimilarity = 53.45%	
Av. Diss.	Diss./SD	Contrib.%	
Siganus luridus	6.52	1.11	12.20	12.20	
Sarpa salpa	5.95	1.26	11.13	23.32	
Oblada melanura	5.57	1.26	10.42	33.75	
Diplodus vulgaris	4.89	1.24	9.15	42.90	
Sparisoma cretense	4.60	1.26	8.60	51.49	
Serranus scriba	3.64	1.24	6.81	58.30	
Scorpaena spp.	3.22	0.92	6.02	64.32	
Diplodus annularis	3.06	1.11	5.72	70.04	
Diplodus sargus	2.99	1.32	5.59	75.63	
Siganus rivulatus	2.85	0.63	5.34	80.97	
Serranus cabrilla	2.79	1.11	5.21	86.18	
Mullus surmuletus	2.14	1.12	4.01	90.19	
Note:

Av. Diss., average dissimilarity; Diss./SD, dissimilarity to standard deviation ratio; Contrib.%, percentage contribution of the different species to the overall dissimilarity; Cum.%, cumulative percentage contribution of the different species to the overall dissimilarity.

Discussion

Statistically significant differences were detected between line and strip transects in the estimates of occupancy and species richness. We suggest that the higher overall estimates of occupancy and species richness by the line transect method are mainly due to the greater width of the line transects, and thus the larger area surveyed. The use of narrow strips is dictated by the need to satisfy the assumption of perfect detectability, which is the main assumption of strip transects (Katsanevakis et al., 2012). On the contrary, in line transects perfect detection is required only “on the line”; this allows expanding the width of the transects and increases the probability of recording less common species. Furthermore, the reaction of fish to the presence of the observer can be crucial for the detection of a species. Many shy species may react to divers by fleeing at distances greater than the fixed width of the strip transect, before being detected by the observers at their initial position, and hence remain unrecorded. Bozec et al. (2011) indicated that shy species display clear avoidance behavior toward divers, while the distance from the observer increases with fish size. The appropriate width of the strip transect to ensure species detection may differ even for closely related species (Kulbicki & Sarramégna, 1999), or even for the same species in a different habitat (Smith & Nydegger, 1985; Ensign, Angermeier & Dolloff, 1995; Cheal & Thompson, 1997). By extending the surveyed width through the use of line transects (if there are no other limitations), these sources of error can be reduced.

With regard to overall fish density, line transects again led to a higher estimate than strip transects. This difference is partly related to fish behavior (Bozec et al., 2011; Pais & Cabral, 2017). Kulbicki (1998) pointed out that fish are not motionless items and, in most cases, will either avoid or be attracted to an observer, with the reaction sometimes changing from site to site. The frequency of shy species peaks at intermediate distances because they tend to keep a distance from the observer. The frequency distribution of distances for the majority of the species in the present study followed the pattern of “shy” species. In these cases, the peak of the distance frequency distribution of fish observations was at distances between 0.7 and 2.2 m from the line. As this peak reflects the combination of flee behavior and declining detectability with distance, it is quite probable that many individuals had fled at distances much greater than the observed peak. Fish behavior is therefore, a possible reason why line transects, which utilized a wider surface area (i.e., 8 m on either side of the transect), yielded higher overall density estimates compared to strip transects (i.e., 2.5 m on either side of the transect), because some shy fish could have moved beyond the limit of the transects and thus were not recorded. Nevertheless, many individuals seemed to flee at distances <2.5 m and thus would have been included in the strip transects.

Another important factor which may lead to a potential underestimation of abundance in strip transects, especially for secretive species, is imperfect detectability (Franzreb, 1981; Kulbicki, 1998). The results from DISTANCE analysis showed that a sharp decline in detectability is obvious at distances >2.5 m from the transect line for the majority of the surveyed species. Moreover, numerous studies have also shown that an obvious decline in detectability is observed at approximately 3 m distance from the transect (Harmelin-Vivien et al., 1985; Smith & Nydegger, 1985; Fowler, 1987; McCormick & Choat, 1987; Cheal & Thompson, 1997; Kulbicki & Sarramégna, 1999), although this can be case-dependent. According to the above, the 2.5 m width on each side of the strip transects used in the present study should be sufficient in many cases for the detection of the majority of the target species. However, there were several exceptions, such as Scorpaena spp. (Fig. 2), S. cabrilla, and S. scriba, which presented a substantial decline in detectability at distances <2.5 m. For these latter species the density estimates by line transects were substantially higher than by strip transects.

Although in most cases line transects yielded higher estimates, this method has additional potential sources of bias. An important assumption in distance methodology is that fish should be recorded prior to any movement in response to the observer. Violation of this assumption leads to a negative bias in abundance estimates of “shy” species (Buckland et al., 1993). Moreover, the additional time needed to carry out the distance measurements and the actual deployment of a tape-measure, may further augment the fleeing response of more mobile fish, and hence lead to an underestimation of their numbers during line transects. This source of bias is considered to be more intense in areas of high fish densities, since the additional time needed for measuring the distance would in this case cause more fish to be undetected (Watson, Carlos & Samoylis, 1995). Finally, transects are not snapshots of species distribution and thus new individuals may enter the sampling area during a survey, causing an, overestimation of fish density. This has been recently demonstrated through spatially-explicit individual-based models of fish movement (Ward-Paige, Flemming & Lotze, 2010; Pais & Cabral, 2017). As line transects take more time than strip transects for the same distance, at slower average speed, fish movement per se (i.e., not in response to the observers) may lead to higher counts for mobile species.

The multivariate analysis of the species composition indicated an overall high resemblance between the two methods. In most stations the majority of the species recorded by one method were also recorded by the other method, and at similar densities. These results suggest that the choice of a specific method (either plot sampling or distance sampling) should not significantly affect the overall outcome regarding the spatial patterns of species composition, especially in large-scale studies.

Unfortunately, as is the case in most field studies, the real density values of the fish species in the areas under study were not known. Therefore, it is not easy to determine which is the “best” method by providing precise estimates of the biases related to each method per species. According to several studies, distance sampling appears to be advantageous in many cases. Kulbicki & Sarramégna (1999) have proposed that the use of distance sampling method in UVS could potentially improve estimates by yielding values closer to the true values. Similarly, Ensign, Angermeier & Dolloff (1995) showed that distance sampling, compared to quadrat sampling and strip transects, produced density estimates that were closer to true densities, while Thresher & Gunn (1986) proposed that distance sampling should be preferred for the assessment of secretive species. Irigoyen et al. (2018) also proposed the distance sampling as an appropriate method to survey medium- and large-sized fish species, although they also discussed some disadvantages of the specific method that should be taken into consideration.

Conclusion

Both methods have several specific advantages and limitations, and both are prone to biases. Strip transects suffer from imperfect detectability and the related necessity of narrow transect widths, which may cause underestimation of densities, occupancy, and species richness. In line transect sampling, detection probability is properly taken into account, but the assumption that all individuals are detected at their initial position is difficult to satisfy, especially for fish of high mobility. Line transect sampling is expected to provide much more accurate estimates than strip transect sampling in the case of secretive species of low mobility. An additional advantage of the line transect method is that it provides a way to assess fish behavior through the analysis of distance frequency graphs. On the other hand, in the case of mobile species with neutral or close to neutral behavior, and especially at high fish densities, strip transects would probably be more efficient, as line transects are time-consuming and the disturbance of fish would be higher due to the distance measurements. The choice of the best method to apply needs careful consideration and depends on the aims of each study, the target species, and the peculiarities of the study area. Joint application of both methods could be considered, with line transects applied by one observer for secretive and large fish, and strip transects by another observer for the bulk of medium-sized mobile fish. Other approaches have been proposed when targeting multiple species with varying behaviors, such as strip transects of various sizes depending on the size and behavior of species (Minte-Vera, de Moura & Francini-Filho, 2008; Prato et al., 2017) or the post hoc use of correction factors for each species, estimated by models, to account for behavioral patterns (assuming their consistency and replicability) (Pais & Cabral, 2017). Further research is needed to improve the performance of line transects and strip transects and reduce their biases, as well as to compare the various proposed approaches and field protocols when targeting multiple species with varying behaviors.

Supplemental Information

Supplemental Information 1 Raw data of both distance and plot sampling fish recordings.

Click here for additional data file.

We thank the following diving centers for helping to carry out the fieldwork: Aquacore Divers, Athos Scuba Diving Center, Azure Diving Center, Lesvos Scuba Oceanic Center, Tortuga Diving Center Mesta Chios, Mystic Blue Eco sailing, and Diving.

Additional Information and Declarations

Competing Interests

Author Contributions

Data Availability

The authors declare that they have no competing interests.

Zoi Thanopoulou conceived and designed the experiments, performed the experiments, analyzed the data, contributed reagents/materials/analysis tools, prepared figures and/or tables, authored or reviewed drafts of the paper, approved the final draft.

Maria Sini conceived and designed the experiments, performed the experiments, analyzed the data, contributed reagents/materials/analysis tools, prepared figures and/or tables, authored or reviewed drafts of the paper, approved the final draft.

Konstantinos Vatikiotis performed the experiments, analyzed the data, contributed reagents/materials/analysis tools, authored or reviewed drafts of the paper, approved the final draft.

Christos Katsoupis performed the experiments, analyzed the data, contributed reagents/materials/analysis tools, authored or reviewed drafts of the paper, approved the final draft.

Panayiotis G. Dimitrakopoulos conceived and designed the experiments, contributed reagents/materials/analysis tools, authored or reviewed drafts of the paper, approved the final draft.

Stelios Katsanevakis conceived and designed the experiments, analyzed the data, contributed reagents/materials/analysis tools, prepared figures and/or tables, authored or reviewed drafts of the paper, approved the final draft.

The following information was supplied regarding data availability:

The raw data are provided in a Supplemental File.

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
