# Peer review of "How many fish? Comparison of two underwater visual sampling methods for monitoring fish communities"

_PeerJ, doi:10.7717/peerj.5066_

## Round 0.1 · original submission · Major Revisions

I apologize for the delay in completing my decision on your manuscript. I had nearly finished my review when my computer developed a problem and it took a few days before I was able to access my completed work.

We received two excellent reviews of your manuscript with numerous, detailed, helpful suggestions. Both reviewers indicate that this is a publishable study but that it requires some revisions. I concur with this view and have some supplementary comments of my own. In addition, I have provided an annotated pdf with minor grammatical and word-use corrections as well as suggestions for a more concise presentation. I highlighted sections that I thought needed change and used inserted comments to suggest alternative wording or changes needed (e.g. delete, add comma, etc.).

In your response to the reviews, you should respond to each of the reviewer's comments as well as my comments below. You don't need to respond to all the minor comments on the annotated version unless you disagree with a proposed change in which case you should explain your reason.

If any of my comments are unclear, you can feel free to email me directly before finishing your revisions.

Editor's comments

While the Introduction and Discussion are generally sound, what seems to be missing is a clear indication of the knowledge gap addressed by your study. The Introduction should explicitly indicate what studies previously compared these or similar methods and briefly indicate their general findings and any problems. In other words, it should be clear to the reader why your study was needed. In the Discussion, you should explicitly compare your findings, qualitatively and quantitatively if possible, to previous comparisons of these methods.

As also noted by the reviewers, more detail on your survey methods is needed for the reproducibility of your study. The only place you mention that the survey was done by SCUBA is the Abstract. This should be specified in Methods. The number of divers and their relative positions as well as swimming speed seem important as is the range in time of day of the surveys. Did the same diver(s) do all the counts? Was there any specific training or prior exposure to the Method or a test of consistency between observers if they differed? How exactly did you measure the lateral distance to fish in the line method? Most fish won't wait for you to swim up to them, so you must have had a way of determining the position and some estimate of the accuracy of this process. Did the sites have similar status as reserves or locations open to spear-fishing as this can make a difference in fish avoidance of divers? Consider whether there is anything else a researcher trying to repeat your approach would need to know to repeat or interpret your study, quantitatively as well as qualitatively.

The reviewers also indicate that more information is needed on placement of the transects. I presume that the strip and line transects did not overlap spatially, but this is not completely clear. Was there a minimum distance between transects within stations and between stations within sites? Precisely how did you randomly locate transect positions. For example, did this involve randomly selecting both starting point and direction? How did this vary within and between stations and sites? How did you define the set of positions from which the actual transects were randomly chosen? Did you have a protocol that specified which method would be used first? Were all transects at a given site done on the same day?

Although the reviewers did not raise many questions about the statistical analysis, some aspects of your approach were not clear to me. Although you may be correct, you may need to justify your approach to readers, like me, with only moderate statistical knowledge.
" For occupancy, your measure is the percent (out of a total of 61 stations) that had each of 20 species. It seems that a straightforward analysis would be a paired comparison (with species as the units, n = 20), parametric if the data are normally distributed and nonparametric if not normal, of the number of occupied stations by each method. I suspect that number of stations is more appropriate than percent. Why did you bootstrap an estimate of the difference in percent to see if it is significantly different from zero?
" For richness, your measure is the number of species (out of a total of 20) that were present at each of 61 stations. Again, it seems that a paired comparison of richness values for each method (paired by station) would be straightforward rather than bootstrapping a difference between methods. In addition, you need to include the actual values of species richness in the text. Both text and Fig. 4 only address differences. Be clear in presenting the data that you indicate that the maximum is 20 as you are only considering a limited set.
" For overall density, your data set is similar to that of richness, and the same question regarding estimating the difference rather than comparing the means with a paired test applies. In addition, in calculating means for each species, you excluded stations in which the species did not occur. I assume from this that if a species was recorded by one method and not by the other, you considered a species present at a station and the zero density was recorded for one method. If that is correct, it might be good to make it a bit more explicit in your Methods. If that is not correct, additional clarification is needed. When you present the data in the Results, it is important that you clarify that this density estimate is a density where the species where the species was found for the 20 study species. In addition, you should give the values for each method, not just the difference.
" For the individual species densities, the issues mentioned in the previous point apply. In addition, it is important that somewhere in your data you indicate what the sample size was for each density calculation (i.e., number of stations where the species was present).
" Where you use parametric measures of distribution, you should indicate (in methods) that the data are normally distributed. Otherwise, you should use non-parametric measures. In addition, SD not SE is considered the appropriate measure of variation in observational data.

The rationale for the order of your list of species (Table 1) is not evident. I would have expected species in the same family to be grouped together and the order of families to match standard ichthyological sources. In addition, the species order should remain consistent between tables and figures, which is not presently the case. Can you specify in the table or caption or text the species that might be included in Scorpaena?

You often refer to fish avoiding an observer but don't link your study to the established terminology such as 'flight initiation distance (FID)' or 'flight distance'. This concept might be helpful to your explanation and there are several studies that relate it to validity of counts, for example Feary et al. 2011 Conservation Biology 25:341-349. However, I will not insist on adding this literature.

For your key words, you should consider alternative terms for your methods that are used by other researchers so that such researchers can find the article more easily (e.g., belt transect is sometimes used instead of strip transect and there are, no doubt, others. It is more important to include terms that are not in the title or abstract where they are picked out more readily by searching algorithms. Consider adding Mediterranean because that is for that reason.

Table 2. In the heading, a should be a subscript of P, and the asterisk should be explained.

L224 and Fig. 2. There is some lack of clarity and accurate citation in your presentation of fish characteristics. First, crypticity is not really a behavior per se, although it does depend on behavior as well as coloration and pattern. Second, Kubicki 1998 used the term secretive rather than cryptic. In his 2011 paper in Marine Biology with Bozec as first author, he did use the term cryptic, but he separated fish characteristics into shyness (3 categories) and crypticity (2 categories). This makes more sense to me.

The caption of Fig. 2 is not clear or complete. The expression 'typical distance distributions' seems to imply that the data distribution may not only a sample for those species. It would make more sense to be more explicit. In addition, I assume that the bars are not detection probability values but are something like the number or proportion of individuals detected in different distance categories. They also need to be described. I don't see how detection probability can be greater than one, which matches the line but not the bars, so the ordinate label may be misleading. Shouldn't the probability ordinate reach a logical maximum at 1.0 with a second ordinate for the bars on the right side of the figures? If you cite Kulbicki in the text, I don't think you need to cite him in the caption. If you retain different scales in the three panels, alert the reader to that fact in the caption. A caption, adding in missing information, could be something like 'Detection probability curves (lines, left ordinate) calculated for three species recorded in line transects illustrating (A) crypticity (Scorpaena spp.), (B) shy behavior (Diplodus annularis) and (C) neutral behavior (Epinephelus costae). Bars (right ordinate) represent the number (proportion?) of observations in 1-m intervals from the line.' If the bars are proportions, you also need to give the total number of observations for each species; it might be a good idea to do so anyway. Do you think you should mention in the text if you did not find any species that were attracted to the divers?

Fig. 3. The caption is incomplete. Because this is your presentation of occupancy data, the word 'occupancy' should be prominent. The color coding for each method should be included in the caption (and also the position in case someone does not have a colored version). Indicate the total number of stations. (Most of this could easily be added in parentheses after the relevant words of the present caption.) Explicitly state what the numbers at the ends of the bars represent. Also, it would be useful to present two bars at the bottom representing the mean occupancy because this is what your analysis compared. Keep the species in the same order as in other tables and figures.

Fig. 4. The caption should include the calculation of the difference (which method was first) so that differences are meaningful to the reader.

Tables 3 and 4 and Fig. 5 overlap excessively. My feeling is that the tables convey the density information more clearly than the figure does and that it would be quite easy to combine the actual density with the difference to make a single table and remove the figure. Using landscape format will provide space for more columns. You can include overall density for both measures, but I think it makes more sense as the last row than the first one. You can then refer to it instead of having the data in the text. Using 'Mean density (no./ha)' as the column head will save space (explaining what your density measure actually means in the heading). Note that the abbreviation for number is no. and for hectare is ha. Putting the CI as a range (72.5 - 132.7) with a column head '95% CI (no./ha)' will also save space. You can use an asterisk system to indicate significant differences, if specified in the caption. Also note that the means should be aligned by decimals and that all values should have the same decimal precision. You can add a column for the number of stations included. Is the density of zero in all rows for D. labrax consistent not estimating density from sites where the species was absent? Is an explanation in the heading needed? In addition, there is an incorrect abbreviation of Epinephalus (should be E.) and capitalization of the species name. Finally, while you are working on the table, you might as well remove the vertical lines which are almost never allowed in tables in refereed journals.

If you do decide to retain Fig. 5, it will be necessary to change the symbols so that they will be clear to someone who has a copy in black and white. Also, for descriptive data, SD not SE is usually the appropriate measure (or a non-parametric measure such as quartiles if the data are not normally distributed).

Fig. 6. Caption should explain what the lines are.

The species composition paragraph (L268ff) is a bit too brief to be easily understood. For example, when you provide station numbers that showed high and low similarity, you should state in the text (and caption) that these points are indicated by circles and arrows, respectively. You should also indicate the relevant station numbers for the North and South Aegean locations.

L284-285. The verb 'are mainly attributed' is ambiguous because of the passive voice and lack of a clear agent. Do you mean that you suggest this explanation? If so, you might write something like 'We suggest that the higher overall estimates . . . are mainly due to . . .' I also proposed some changes in the text for clarity and grammar. If it is someone else who is attributing, specify the reference.

L305-310. Reviewer 1 questions the logic of arguing the line transects include fish that might have moved out of a strip transect. He points out that divers are usually looking ahead to count fish before they move and that the peak of frequency for line transects is often within the distance of the edges of strip transects. I can see that if peak is within a strip transect width, there is still an implication that some fish are fleeing at greater distances, but I think you need to develop the logic of your argument more completely.

L336. Expand the sentence to briefly indicate why the bias is greater at higher density.

L347. Reviewer 1 asks for references, but my understanding is that the relevant references are incorporated into the next two sentences. If so, repeating the references is not required.

·

Basic reporting

The manuscript presents data from two different underwater sampling methods (strip transects and line transects) on rocky reefs in the Aegean Sea to investigate the performance of each in documenting associated fishes. I commend the authors on sampling a larger number of stations, however the manuscript could benefit from some additional information, clarity in the methods, and interpretation of the results.

1. Line 39 This sentence would read better if “last” was replaced by past.

2. Line 40 I suggest re-wording this sentence for clarity, e.g., Selecting the most suitable sampling method…

3. Line 46 The majority of relevant literature uses the abbreviation “UVC” (underwater visual census) instead of UVS. In fact, you use UVC in line 349. It might be worth adopting this abbreviation for consistency.

4. Lines 39 – 52 This study deals with two different underwater visual assessment methods. I would suggest keeping the focus here on underwater methods rather than discussing non-observational techniques. For example, the five techniques in paragraph two could be discussed in more detail (pros and cons).

5. Line 58 The authors need to explain why these two methods were selected over the other three for this study.

6. Line 70 While this might be an assumption, I don’t think anyone really assumes this is the case in reality. Clearly there are species that are difficult to detect as you have pointed out in the subsequent sentences. I suggest that you provide a reference to support this statement and maybe remove the word critical.
Most studies assume that sampling is not perfect. Replication, transect parameters, and survey method is therefore selected to maximize accuracy and increase the probability of recording the majority of the individuals in the sampling area. Furthermore, it would appear that most of these factors that influence fish detection discussed in the subsequent sentences for strip transects would also apply for line transects.

7. Line 94 Is there an accepted level of precision? e.g., nearest mm, cm, 5cm, etc.,?

8. Line 99 The authors need to discuss the tradeoffs associated with both methods and when each method would be preferred over the other.

9. Line 114 More detail is needed here on the specific aims of the study. E.g., what metrics are being tested? Also, you have a pre-selected group of fish in your study. Is this specific to reef health, fisheries, or another reason?

10. Line 122 How were the stations independent of each other within each site? They appear to be two site-level replicates as there was no replication at a transect level.

11. How were the sites selected and was there any criteria for selection that is important to the study?

12. Line 127 suggest replacing “at a” with “where”
and “off” with “was”

13. Line 133 and 133 Replace “cluster” with “school”

14. The authors need to give more detail on how the two methods were conducted within each station. How far apart were the transects, was there the possibility of one survey influencing the other, was the habitat similar between the two, etc.,

15. Is there a reason why these twenty fish were selected? Please note this.

16. For a reader that is not familiar with the “line method” it would be useful to provide some additional information on the process of conducting this technique. How does the observer use a measuring tape to estimate the distance for one fish without influencing nearby fish behavior? How does the observer deal with mobile fish?

17. Line 220 What are the variance estimates for each of the values? Is site a factor in the model? I would assume that variables associated with each site (habitat complexity, water visibility) would influence the detectability?

18. Line 220 Why are there not figures for all species (Figure 2)?

19. Line 225 – 229. Can you confidently state that detectability for each of your species is dictated by active avoidance? Could this be linked to feeding behaviour, mobility, etc.,

20. Line 232 Was D. vulgaris observed in both methods within the same sites? For Figure 3, it would be informative to know if occupancy was similar between methods within a site.

21. For clarity, use “species” instead of “cases”

22. Line 249 It is unclear why you have presented two tables (Table 3 and 4) and a figure (Figure 5) for fish density, and why you have presented CI on one and SE on the other.

23. Line 269 How were the stations selected for analysis inclusion? Line 271 It would be useful to include this into the supplementary information. While the figure may benefit from less stations, all stations should be included into a formal analysis.

24. Line 272 It is unclear how you came to the conclusion that the two methods present similar species composition. What analysis supports this statement? The study should include some analysis to support these statements (e.g., ANOSIM).

25. Line 277 This result appears out of context as it is not related to the aims of the study. I suggest that you remove this.

26. Line 285 This suggest that a width of 5m provides perfect detectability. Obviously this would vary depending on the species of fish and water visibility, and not the fish community.

27. Line 288 It is unclear what “on the line” means

28. Lines 289 – 292 While shy fish may flee the observer, if they were originally within the bounds of the transect parameters then shouldn’t they still be counted? It shouldn’t matter that they then swam outside the transect or that they fled a greater distance. Fish observers are generally looking and recording fish ahead of them within the transect, and not directly to the sides of them. In addition, I would assume that a species reaction to an observer would be consistent for both methods?

29. Line 297 transect widths are often dictated by the targeted fish species (size, behaviour, cryptic nature) and environmental variables. Often the width of the transect cannot just be extended. Factors like water visibility and habitat architecture can mean that transect widths often need to be kept narrow.

30. Lines 300 – 306 I would assume that fish behavior and shyness is consistent among methods?

31. Line 303 “>0” this is not very informative

32. Lines 306 – 311 This appears to contradict itself. Especially as the peak (0.7 – 2.2m) is within the strip transect dimensions

33. Lines 329 – 337 An important aspect that needs to be stressed for surveying fish is time. Time equals money, but is also a significant limitation when using scuba. This would result in less bottom time, which would reduce the number of sites/replicates. What is the difference in sampling time for the two methods?

34. Lines 347 Please provide some references to support this statement

35. Lines 344 – 353 There are many tradeoffs associated with this. It would be interesting to know how long it took to complete each method as this is always a concern for someone designing a sampling protocol.

36. Line 371 this is repetitive

37. Line 373 In addition, a single strip transect can be conducted with two different widths to account for different fish species. A larger width for larger, mobile, shy species, and a narrower width for small, benthic, cryptic fishes.

Experimental design

38. There is no replication at a station level. There is only one transect at each station for each method

39. It appears that a large proportion of the differences between the two methods could be attributed to the different sampling areas (see lines 283 – 285, 306, 312 - 328). Strip transects had a width of 5m (area 375 m2) while line transects has a width of 16m (area 1200 m2). A standardised sampling area for both methods would have reduced the bias between the two. While I appreciate that the sampling cannot be repeated, it might be useful to test the proportion of variance attributed to sampling area.

40. It appears that this study pooled all the stations from each of the two methods together to make the comparisons. To accurately assess the two methods I would assume that a pair-wise design would be more suited and account for differences among sites.

41. Some basic statistics are missing from the results. E.g., line 250, 258, and for community comparisons (nMDS). Line 274, the authors could use a SIMPER analysis to show what species are resulting in the observed difference. This could then be linked back to the distance sampling results. The authors should provide support for these statements.

Validity of the findings

42. It is unclear how much of the variation between the two methods is driven by the actual observational methods or the difference in the surveyed area (see comment 39). In addition, factors like time need to be considered in order to make a case that one method is preferable to another. The conclusions appear to suggest that two methods suit different groups of fishes (cryptic vs conspicuous) see lines 33 – 35, however it appear that most of the species in this study could not be categorized as cryptic. The authors should recommend the method that suits the fish group that was targeted in this study best.

Additional comments

43. Overall, more information is needed in the table and figure captions.
In addition, Figure 6 could be labeled differently to better illustrate the data. Many of the symbols look similar. E.g., instead of symbols on the plot, just have the site number. These can be colour coded to represent the observation method. It is also not obvious what the lines and circles represent.

·

Basic reporting

The manuscript overall is clear. English is good in general, except for the first line of the abstract where “are preferred than” should be “are preferred over”.

L61: This statement refers to the present use of UVS, and yet it cites papers from the 90s. Consider citing more recent work, such as Caldwell et al. (PLoS ONE 2016) and/or others.

L89: “a” should be subscript in “Pa”

Introduction is well structured and provides good background information. However, I was surprised to see no reference to previous comparisons of line and strip transects, such as the (rather old but relevant) papers by Burnham and colleagues in 1984 and 1985 (J Wildlife Management).

One component of bias that is not mentioned is the fact that transects (strip or line) are not snapshots. The way we calculate densities for both methods assumes that no new fish entered the sampling area as we sampled, but that is not the case. It is not only about fish movement due to the observer, but also about fish movement per se. And the faster the fish move relative to the observer, the highest the potential overestimation this may cause. This was discussed by Lincoln Smith (MEPS 1988), and more recently observed in simulation approaches such as Ward-Paige et al (PLoS ONE 2010) or Pais and Cabral (Ecological Modelling 2017) (already cited, not shamelessly asking for citations here). This is relevant because you refer line transects take more time for the same distance, so I assume an average slower speed. This could, at least theoretically, lead to higher counts for mobile species.

Tables and figures:
Figures in general are quite heterogeneous. They seem to be formatted exactly as they are output by the different software packages. Consider matching figure styles where appropriate, at least for the same type of graph (e.g. histograms).

Please correct typos in species names (D. labrax on all tables, E. marginatus on t3).

Table 2: asterisks should be explained in the legend or as a footnote.

Table 3 is very similar to figure 5 and more difficult to interpret. Maybe just a figure 5 with 95% CI instead of standard errors could be a hybrid solution. If error bars are too big, then standard errors would suffice, and maybe table 3 could go as supplemental data.

Table 4 could also benefit from being represented as points and error bars, with zero as a vertical line separating species that are favoured by either line or strip transects. This would greatly improve interpretation and the values in table 4 could still be supplemental.

Figure 3, on the other hand, simply compares 2 values per species, so it could be easily interpreted on a table. Simply marking the highest value in bold on a table would help interpret the pattern.

Experimental design

Some aspects in the methods section should be clearly stated to help with replication:

- the average swim speed (or time taken) for each transect type is very important, since it is known to affect bias.

- The number of transects per station should be clear. This is important because between-transect variability can be quite large, and even two transects on the same site at different times can lead to very different density/richness estimates. If sampling effort is small, or very different sites are being pooled together, this should be discussed, and any possible implications on the interpretation of results.

The explanation for picking one of the two stations from each site should be on the methods (2.5) and not on the results (3.5), as it is not clear initially how and why this was done. You could still mention in the results that picking the other stations led to similar results. An alternative would be to aggregate both stations per site into a single point, so the choice of picking one could be addressed in the discussion.

Validity of the findings

Data analysis and results seem adequate, but see concerns with replicability and replication in point 2 above.

L299 onwards: Other authors (e.g. Sale and Sharp 1983) observed that wider transects lead to lower density estimates. This means that it is probably not just width that explains higher densities in line transects. I suspect it also has something to do with additional, non-counted fish being added to line transects to compensate for detectability, something that is not done in strip transects.

It would be useful to discuss what your findings mean to the viability of other proposed methods such as nested sampling (e.g. Prato et al. 2017 PLoS ONE, Minte-Vera et al. 2008 MEPS), or the application of (constant) correction factors per species to strip transects (e.g. Christensen and Winterbottom 1981, Ward-Paige et al. 2010, Pais and Cabral 2017, etc.)

Additional comments

The paper is well structured and approaches important research questions about the choice of UVS method, while providing important data on species behaviour and detectability for the study area. The authors address density estimates on a per-species basis, which I believe it is the correct approach, and they do not fall into the temptation of referring to higher estimates as better, as it has often been done in the literature. This is even specifically referred in the discussion, and gladly so.

The paper fits the scope of the journal and I believe it can be published if some concerns are addressed, particularly those related to reproducibility, presentation and discussion of results. I enjoyed reading the paper and most of my comments are suggestions that can hopefully help improve the manuscript.

---

## Round 0.2 · accepted · Accept

Thank you for your thorough and careful revision and response to the reviewer and editor comments. The manuscript is now ready for publication. I have attached a pdf with a few minor suggestions for changes in wording or elimination of redundancies that can be addressed by PeerJ staff during production.

#